# Invasive Pneumococcal Diseases in People over 65 in Veneto Region Surveillance

**DOI:** 10.3390/vaccines12111202

**Published:** 2024-10-23

**Authors:** Silvia Cocchio, Claudia Cozzolino, Andrea Cozza, Patrizia Furlan, Irene Amoruso, Francesca Zanella, Filippo Da Re, Debora Ballarin, Gloria Pagin, Davide Gentili, Michele Tonon, Francesca Russo, Tatjana Baldovin, Vincenzo Baldo

**Affiliations:** 1Department of Cardiac, Thoracic, Vascular Sciences, and Public Health, University of Padua, 35128 Padua, Italy; silvia.cocchio@unipd.it (S.C.); claudia.cozzolino@phd.unipd.it (C.C.); andrea.cozza@studenti.unipd.it (A.C.); patrizia.furlan@unipd.it (P.F.); irene.amoruso@unipd.it (I.A.); tatjana.baldovin@unipd.it (T.B.); 2Preventive Medicine and Risk Assessment Unit, Azienda Ospedale Università Padova, 35128 Padua, Italy; 3Regional Directorate of Prevention, Food Safety, Veterinary, Public Health—Veneto Region, 30123 Venice, Italy; francesca.zanella@regione.veneto.it (F.Z.); filippo.dare@regione.veneto.it (F.D.R.); debora.ballarin@regione.veneto.it (D.B.); gloria.pagin@regione.veneto.it (G.P.); davide.gentili@regione.veneto.it (D.G.); francesca.russo@regione.veneto.it (F.R.)

**Keywords:** invasive bacterial diseases, *Streptococcus pneumoniae*, elderly, surveillance, pneumococcal vaccination

## Abstract

Background: Elderly individuals over 65, along with children under 5, are the most affected by invasive pneumococcal diseases (IPDs). Monitoring vaccination coverage and conducting surveillance are essential for guiding evidence-based prevention campaigns and public health measures. Methods: Since 2007, the Veneto Region has relied on three sources for surveillance of invasive bacterial infections, contributing to an increase in reported IPD cases. This study analyzed notifications related to individuals aged ≥65 years from 2007 to 2023. Results: A total of 1527 cases of IPDs in elderly individuals were reported between 2007 and 2023. The notification rate significantly increased from 5.61 to 14.63 per 100,000 inhabitants, despite underreporting during the COVID-19 pandemic. Cases associated with sepsis increased from 3.89 to 9.58 per 100,000, while notifications of meningitis and case fatality rates remained stable at 1.5 per 100,000 and 11.8%, respectively. Serotyping was not performed in 52% of the notifications. The most common serotypes were 3 (21.6%), 8 (11.1%), and 19A (5.0%), with fluctuations over time. There was a significant decline in serotypes covered by PCV7 and PCV13 and an increase in non-vaccine serotypes. Conclusions: The regional surveillance system allows for an increasingly comprehensive profile of the epidemiological landscape of IPDs in Veneto. However, the surveillance of pneumococcal infections still presents challenges. The currently available data are likely to be underestimated, mainly referring to the most severe cases, and the serotyping necessary to identify the etiological agent is still not often performed.

## 1. Introduction

*Streptococcus pneumoniae* (pneumococcus) is a common cause of infection, especially for children and the elderly, sometimes with very serious complications and outcomes. This pathogen can, in fact, most frequently be spread by an asymptomatic carrier or, to a lesser extent, by a person with a pneumococcal infection. By inhalation, the pneumococcus colonizes mainly in the nasopharynx. It can lead to non-invasive pneumococcal diseases such as sinusitis, otitis media, and (non-bacterial) pneumonia. A pneumococcal infection can also lead to invasive pneumococcal diseases such as bacteremia, meningitis, and bacteremic pneumonia and also endocarditis, septic arthritis, and peritonitis [1,2]. In severe cases, it can lead to the death of the subject. Invasive pneumococcal diseases (IPDs) are an important part of invasive bacterial diseases (IBDs) together with meningococcal and *Haemophilus influenzae*-related diseases.

According to the latest ECDC report for 2018, there were 24,663 confirmed cases of IPD [3] in Europe and the European Economic Area. Most of the cases involved children under one year of age (14.4 confirmed cases per 100,000 population) and adults over 65 years of age (18.7 per 100,000 population) [3].

The trend appears to be seasonal with the greatest number of cases attributable to the fall and winter periods [3]. Since 2014, there has also been an upward trend in confirmed cases of IPD going from 17,528 cases in 2014 to the aforementioned 24,663 confirmed cases in 2018, with a year-on-year increase over the period [3].

Pneumococcus can be distinguished into more than 90 different serotypes depending on the capsule polysaccharide complex. Among the various serotypes, those responsible for severe disease are serotypes 3, 4, 6B, 9V, 14, 18C, 19F, and 23F. They are responsible for about 60 percent of invasive disease in the population and 90 percent of invasive disease in children [2]. Serotype 3 is responsible for the majority of invasive pneumococcal disease in the adult population [2]. At the European level, the most represented serotypes in 2018 were, in descending order of frequency, 8, 3, 19A, 22F, 12F, 9N, 15A, 10A, 23B, 6C, and 11A [3]. These accounted for 70% of the serotypes found among those typed [3].

Currently, two types of vaccines are available for the prevention of pneumococcal-related diseases: the polysaccharide pneumococcal vaccine (PPSV) and the conjugate pneumococcal vaccine (PCV) [4]. These vaccines are used differently across countries, depending on national immunization guidelines and the target population, whether children or adults. The conjugate pneumococcal vaccine provides coverage for up to 21 strains, with the 21-valent version approved by the U.S. Food and Drug Administration (FDA) in June 2024 [5] (not yet available in Europe), while the polysaccharide pneumococcal vaccine offers protection against up to 23 strains [4,6,7].

The latest vaccine calendar of the Veneto Region, in force since summer 2023, provides for the administration of three doses of conjugate vaccine in newborns at the second, fourth, and tenth months of life [8]. In adults, one dose of conjugate vaccine is to be administered in the 65th year of age [8]. In the high-risk elderly population, sequential PCV + PPSV vaccination is also recommended [8]. Anti-pneumococcal vaccination is not mandatory in the Veneto Region.

High-quality surveillance is essential for accurately estimating disease burden [9,10], tracking changes in serotype distribution and antimicrobial resistance, and assessing the impact of vaccination on IPD after vaccine implementation [11]. However, several challenges and factors that bias burden estimates have been identified, including issues with IPD case definition, *S. pneumoniae* laboratory detection, comprehensiveness of surveillance systems, and underreporting [12,13]. Greater standardization and more detailed data reporting could enhance our understanding of pneumococcal epidemiology.

The rationale of this study is to assess the incidence of invasive pneumococcal disease in the over-65 population of the Veneto Region, based on regional surveillance data. We also intended to evaluate how serotypes responsible for IPDs have changed over time in relation to vaccine coverage and valency, particularly focusing on non-vaccine serotype infection rates.

## 2. Materials and Methods

### 2.1. Study Population and Data Source

In this observational study, we analyzed all cases of *Streptococcus pneumoniae* related to IBDs as reported by the surveillance system of the Veneto Region.

Veneto, located in northeastern Italy, is one of the country’s wealthiest regions, with a gross domestic product (GDP) of USD 234,995 million in 2023 [10,14]. It ranks 13th out of Italy’s 20 regions in terms of population age. As of 2022, Veneto had an age index of 195.1, a population density of 264.3 individuals per square kilometer, and an average population of 4.8 million, with a mean age of 46.1 years. Of this population, 50.9% were female [10,15]. The Italian National Health System (Servizio Sanitario Nazionale—SSN, in Italian) operates as a public healthcare system, primarily financed through general taxation and managed at the regional level [10,16].

In the Veneto Region, the surveillance of invasive infections caused by bacterial agents relies on three distinct data collection methods: the Regional Infectious Disease Information System (Sistema Informativo Regionale Malattie Infettive—SIRMI) [17], the National Infectious Disease Reporting System (Sistema di notifica delle malattie infettive PREMAL) [18], and the Special Surveillance for IBDs from the National Institute of Health (Sistema di Sorveglianza Delle Malattie Batteriche Invasive dell’Istituto Superiore di Sanità—ISS) [19]. While the national surveillance system focuses specifically on invasive infections caused by pneumococcus (*Streptococcus pneumoniae*), meningococcus (*Neisseria meningitidis*), and *Haemophilus influenzae*, the Veneto Region’s system includes any bacterial pathogen responsible for infections with invasive characteristics.

Since 2007, regional surveillance has been based on data from these three integrated sources. In 2010, the national flow was consolidated through the implementation of a single computerized system for IBD case recording (Sistema informativo malattie infettive Web—SIMIWEB), which enabled comprehensive and unified management of all notifications. Since 2013, the SIMIWEB system has allowed systematic and concurrent notifications with the regional surveillance system. Finally, as of January 2022, SIMIWEB was replaced by a new regional portal, SIRMI. IBD cases are routinely reported by the Public Health and Hygiene Services of the various regional health units (Azienda Unità Locali Socio-Sanitarie—AULSS) through this platform, as outlined in a Regional Note (no. 145,206 of 30 March 2022).

### 2.2. Inclusion Criteria and Case Definition

This study included all suspected invasive pneumococcal disease diagnoses, which were reported in the Veneto Region surveillance system by the local health authorities and confirmed by the microbiology laboratories from 2007 to 2023. Population included only individuals aged 65 and older. Notifications associated with individuals who were not residing in the Veneto Region were excluded. As previously reported [20], a case of IPD was established from the isolation of *S. pneumoniae* from blood or another normally sterile site, according to the EU definition [21,22].

The pneumococcal isolates sent to the Regional Reference Laboratory were identified using standardized laboratory procedures and, when feasible, serotyped. The identified serotypes were then classified as either vaccine or non-vaccine serotypes (according to Italian vaccination schedule), based on their inclusion in the PCV7, PCV13, PCV15, PCV20, or PPSV23 vaccine (see Appendix B, Table A1). Given the recent U.S. FDA approval of the 21-valent conjugate vaccine [5], we also quantified the cases with serotypes covered by this vaccine, even though it has not yet been approved or made available in our country.

### 2.3. Statistical Analysis

Variables were represented with frequencies and percentages. Pearson’s Chi-squared test, or alternatively Fisher’s exact test, was employed to assess differences in IPD notifications variables among age groups 65–74, 75–84, and over-85.

Annual IPD notification rates per 100,000 were calculated considering the number of confirmed cases divided by the estimated size of the resident population over 65 (source Demo Istat [23]). The case fatality rate (CFR) was calculated by dividing the number of reported deaths by the number of notifications, expressed as a percentage. Notification rates and CFR were further stratified by age groups, sex, serotypes (vaccine and non-vaccine), and clinical manifestations.

Joinpoint regressions [24] were conducted to evaluate the significance of notification rate trends over the years. Results were expressed as annual percentage change (APC) and average APC (AAPC).

Confidence intervals (CIs) were calculated as appropriate. A *p*-value < 0.05 was considered significant for results. All data manipulations, analyses, and visualizations were performed using Python 3.8.18 and R 4.2.2.

### 2.4. Ethics Statement

IPD notification records were obtained from the administrative databases of the Veneto Region, and the disclosure and utilization of such records for educational and scientific purposes did not necessitate approval from ethical committees. On 24 January 2023, the Veneto Region implemented the Code of Conduct for the use of health data for educational and scientific publication purposes (Official Bulletin of the Region, “*Bollettino Ufficiale della Regione*” no. 10), as established by the European Committee (European Regulation 2016/679). This implementation received approval from the Italian Personal Data Protection Authority on 14 January 2021.

Adhering to the current Italian privacy legislation, the publication and utilization of health data, along with the processing methods, must occur exclusively in aggregate form, without any reference to patients’ personal information. Prior to providing access to the authors, all personal data that could potentially lead to identification were substituted with anonymous codes, in accordance with current privacy regulations (Legislative Decree no. 196 of 30 June 2003).

## 3. Results

In total, 1527 cases of invasive infections caused by *S. pneumoniae* involving elderly individuals were reported to the regional surveillance system from 2007 to 2023 (about 54.9% of total all-age IPDs). The majority of patients were male (55.2%) and aged 75–84 years (38.5%). The vaccination status was unknown in more than half of the sample and not available for records prior to 2012, and among those reported, only 29.9% of the elderly were found to be vaccinated. Associated meningitis, sepsis, and other invasive diseases were reported in 17.4%, 62.8%, and 47.1% of the notifications, respectively. Sequelae were observed in 15 subjects (1.2%) following an IPD: 3 auditory, 5 neurological, 1 amputation, and 8 others (with 2 cases with multiple consequences). In 187 reported cases (12.2%), a diagnosis of IPD was followed by death.

More than 95% of the cases were microbiologically confirmed by isolating *S. pneumoniae* from blood.

About 52% of the collected materials were untyped or untypable. The most prevalent *S. pneumoniae* serotypes in the Veneto Region’s IPD cases were 3 (21.6%), 8 (11.1%), 19A (5.0%), 14 (4.1%), and 7F (4.0%). In 29.7% of cases, the isolated serotypes are not covered in any vaccine currently available in the territory, 12.6% are included in PCV7, 47.9% in PCV13, 49.9% in PCV15, 65.9% in PCV20, and 67.5% in PPSV23. Approximately 61.4% of cases had a serotype covered by the latest PCV21 vaccine, but further studies are needed for an accurate understanding of the trend. The characteristics of the sample and the bivariate analysis by age group are shown in Table 1.

Statistically significant differences (*p*-value < 0.05) were found between the age groups. Among reported IPD cases in individuals aged ≥85 years, there were more females (59.4% compared to 39.3% and 41.3% in the 65–74 and 75–84 age groups, respectively), a lower proportion of meningitis (4.6% versus 27.0% and 15.6%), and a higher proportion of other invasive diagnosis (52.8% versus 41.1% and 49.6%). No significant differences in terms of sequelae were observed.

Starting from 2012, vaccination coverage was higher among patients aged 65–84 compared to those over 85 (35.2% and 32.7% versus 16.9%). The observed CFR increased significantly with age, increasing from 9.1% to 12.4% and 17.1%. Liquor samples were significantly more frequent in subjects aged 65–74 compared to older individuals (46.5% versus 29.6% and 11.6%). No significant differences were observed in the distribution of isolated serotypes across age groups.

The invasive pneumococcal disease notification rate ranged from 2.625 to 14.633 cases per 100,000, with the highest value in 2023 and the lowest in 2021.

The data in Figure 1a indicate that overall IPD notification rates showed an increasing trend (AAPC 6.673, 95% CI 3.51; 9.78). The trend was positive from 2007 to 2018 (APC = 6.580, 95% CI 3.41; 11.57), followed by a sharp decline during the pandemic years, 2020–21, (APC 2018–2021 = −35.013, 95% CI −43.38; −20.0). Afterward, a rapid increase was observed (APC 2021–2023 = 125.418, 95% CI 65.77; 193.81, see Appendix A). A similar pattern resulted when stratified by age, with higher, but not statistically significant, rates in the older age groups.

Figure 1b depicts the notification rates of associated specific invasive bacterial diseases. While meningitis remained stable (AAPC 95% CI includes 0), sepsis and other diagnoses generally increased, excluding the COVID-19 pandemic, (AAPC 5.549, 95% CI 2.08; 9.11, and 19.409, 95% CI 7.44; 27.64, respectively).

Although vaccination coverage significantly increased from 17.6% in 2012 to 32.8% in 2023 (AAPC = 8.415, 95% CI 3.54; 13.35), the estimated case fatality rate trend was stationary, oscillating between 6.0% and 16.2% before the coronavirus emergency (see Appendix B, Figure A1).

The typings performed highlight several key trends (see Figure 2a and Table A2). Serotype 3, despite a fluctuating trend, was consistently present throughout the study period, with higher frequencies, both in relative and absolute terms, in recent years, remaining the most represented typed serotype. Serotype 8 has been constantly present since 2009, with a marked increase in frequency since 2015 compared to earlier years. Serotype 19A was consistently present, though not abundant, until 2019, but was only detected again in 2022. Serotype 14 showed a fluctuating presence over the period, with an absence noted in 2010, 2015, 2016, 2017, 2019, 2021, and 2023. Serotype 7F was detected until 2015, after which it disappeared, with a significant reduction in frequency observed in 2014 and 2015 compared to previous years. Serotype 20 showed a fluctuating presence, with relatively low frequency in the years when it was detected. Serotype 6A had a low frequency until 2015, after which it was no longer detected, and similar patterns were seen with serotype 4, serotype 11A, and serotype 23F, all of which disappeared after 2016 or 2015. Serotype 19F showed a variable, sparse presence until 2019, with a low frequency detected in 2023. Other serotypes were variously represented until 2019, with a very low frequency detected in 2022 and 2023.

Stratifying serotypes by vaccination group (Figure 2b), a significantly negative trend appears for those included in PCV7 (AAPC = −12.441, 95% CI −20.13; −4.23) and PCV13 (AAPC = −7.568, 95% CI −13.52; −1.24); for those in PCV15, PCV20, and PPSV23, on the other hand, the trend is stable. The trend of PCV21 shows a similar trend to that of PCV20 and PPSV23, but it is noted that it covers only 61.4% of serotypes typed in the Veneto Region. Instead, the notification trend of non-vaccine serotypes was strongly positive (AAPC 15.322, 95% CI 5.51; 21.65). In addition, although the overall number of notifications reported by the system increased, the portion of isolated typed/typable cases significantly decreased (AAPC untyped 10,017, 95% CI).

## 4. Discussion

This observational study evaluated the trends in invasive bacterial disease cases caused by *S. pneumoniae* in the elderly population of the Veneto Region over the last two decades, using data from the regional surveillance system. Additionally, we aimed to analyze how the distribution of serotypes causing IPDs has evolved over time, particularly in relation to pneumococcal vaccination strategies and campaigns.

Our findings highlight that serotypes 7F, 6A, 4, 11A, and 23F have significantly decreased and have disappeared since 2016 (Figure 2a and Figure A2). Notably, except for serotype 11A, which is included in PCV20 and PPSV23, all the other serotypes have been covered by vaccines in use for many years. This confirms the effectiveness of previous vaccination strategies, consistent with findings from similar Italian studies [25,26].

Conversely, despite being included in PCV13, serotype 3 remains prevalent and consistently ranks as the most frequent serotype causing IPDs throughout the entire study period. Italian national surveillance data also indicate the persistent detection of serotype 3, which continues to be the most widely circulating strain [25,26,27]. This situation can also be explained by considering that this serotype has shown mechanisms of vaccine evasion [28].

Grouping the serotypes of IPDs by vaccine coverage, we observed that 12.6% are included in PCV7, 47.9% in PCV13, 49.9% in PCV15, 65.9% in PCV20, and 67.5% in PPSV23. A multicenter Italian study that assessed the prevalence of community-acquired pneumonia among older adults reported lower percentages [29].

Our surveillance detected a concerning sharp increase in IPD cases with non-vaccine serotypes in this vulnerable age group, as already reported in previous Italian observations [30].

Interestingly, some studies have reported the re-emergence of PPSV23-specific serotypes, in particular serotype 8 [27,30]. In our region, we observed a notable increase in IPD cases involving serotype 8 starting from 2009, increasing from about 4% before 2014 to 16% after 2015, on average (Table A2). A recent review by Teixeira et al. highlights that serotype 8 emerged as the most prevalent cause of IPDs in adults across many European countries, accounting for 15–30% of IPD isolates [31].

ACIP (Advisory Committee on Immunization Practices) findings on the potential of PCV21 [32], which offers coverage for approximately 80% of the circulating serotypes responsible for invasive bacterial disease in the U.S., provide a valuable model for extending and adapting vaccine serotype coverage in Europe. ACIP observations highlight that around 20% of IPD cases are caused by the eight new serotypes included in PCV21. Reformulating immunoprophylaxis strategies to account for both known circulating serotypes and emerging ones may represent a significant advancement in public health approaches.

With regard to the Veneto Region specifically, the available data do not currently allow for a definitive assessment of the benefits from a serotype coverage perspective. Our data indicate that the serotypes covered by PCV21 and that are circulating in the Veneto Region overlap with those already addressed by PCV20 and PPSV23. However, further evaluation would be necessary by extending the typing to untyped serotypes to determine whether they might be covered by the additional serotypes in PCV21.

Given the increasing prevalence of serotype 8 in the Veneto Region, the adoption of vaccines protecting against this serotype seems fundamental.

To prevent infection from this re-emerging serotype, there are some potential vaccination strategies: PCV20 alone, sequential vaccination with PCV15 followed by PPSV23 [33], or the possible subsequent use of PCV21.

Pending the occurrence of any conditions for the introduction of new vaccine technologies, we know that the other two approaches have their advantages and disadvantages. The PCV20 strategy is logistically simpler, requiring only a single vaccination and potentially improving compliance. Further, it has been seen that it is more cost-effective than PCV15 + PPSV23, and if using a single vaccine increases uptake, which is potentially more likely in the underserved population, then PCV20 use becomes even more favorable [34].

In evaluating the best vaccine strategy on the basis of serotype epidemiology, we note that according to the studies by Mt-Isa et al., PCV15 is not inferior to PCV20 in terms of immunogenicity for the serotypes contained in PCV13 [35]. In particular, PCV20 is less immunogenic than PCV15 for certain serotypes, specifically serotype 3, which is the one that is circulating the most in Italy and associated with severe disease [35]. It will be crucial to understand the immunogenicity of next-generation vaccines against serotype 3.

Conversely, expanding vaccine serotype coverage is increasingly important to counteract serotype selection pressure due to the convergent effects of antibiotics and vaccination against *S. pneumoniae* [36]. In addition, it is worth noting that there are increasing rates of severe pneumococcal disease and related mortality [37,38,39,40,41]. Regarding antimicrobial resistance (AMR), extending the coverage of existing vaccines and developing new means are effective strategies to reduce the burden of disease attributable to resistant strains of *S. pneumoniae* [42].

This study has several limitations. First, the data cover only cases from the Veneto Region, which may limit the generalizability of the findings to other populations. Additionally, the dataset is restricted to information available in the notification form, as complete access to patients’ medical histories was not feasible. Vaccination status has only been recorded since 2012, and even when available, it is not consistently documented. We lack comprehensive knowledge about patients’ vaccination status because a patient may not have been vaccinated, and even if vaccinated, the record may not be known to the clinician or included in the medical records. Similarly, follow-up data, including information on sequelae and mortality, are often missing. These gaps restrict our ability to fully assess the long-term outcomes of IPDs and the effectiveness of vaccination. Furthermore, changes in surveillance practices over the study period and potential reporting biases may have influenced the results. While these records are highly specific in identifying severe cases, their sensitivity remains limited. The onset of the COVID-19 pandemic further masked the effects, with a drastic reduction in the number of notifications and likely in the isolation of *S. pneumoniae*, due to shifts in healthcare priorities and reporting.

Surveillance data may thus introduce quantitative distortions in epidemiological estimates and underreporting of both IPD cases and mortality.

## 5. Conclusions

Despite limitations, this study underscores that the surveillance system in the Veneto Region offers an increasingly comprehensive understanding of the epidemiology of invasive pneumococcal diseases. The findings highlight the positive impact of pneumococcal vaccination in both children and adults, with a significant reduction in IPD cases observed among the elderly from 2007 to 2023 for the serotypes covered by the initial vaccines, PCV7 and PCV13. However, there is a noticeable upward trend in notifications for non-vaccine serotypes.

Nevertheless, the regional experience reveals that challenges in monitoring pneumococcal infections persist. In addition to underreporting, our data show that while the overall number of notifications has increased, the proportion of typed isolated cases has declined.

Improving surveillance will hence require continued efforts, not only in raising awareness and ensuring greater engagement among healthcare professionals in accurate case reporting, but also in improving microbiological diagnostic capabilities. More surveillance data from the coming years, along with future research, are needed to better assess the long-term impact of vaccination on the burden of IPDs, particularly with the introduction of PCV15 and PCV20. Additionally, it will be crucial to evaluate the effectiveness of new vaccine technologies and strategies, especially in the elderly population.

Strengthening surveillance should also involve improving both clinical and laboratory monitoring, with attention to non-vaccine serotypes in order to assess future trends.

Finally, regardless of which vaccination strategy is pursued, it is essential to extend vaccination to as many people as possible. Anti-pneumococcal vaccination is also crucial in the pediatric population in order to both protect children and lead to herd immunity [34,43]. Last but not least, considering that pneumococcal vaccination is not mandatory in the Veneto Region, it is strategic to maximize the promotion of adherence to this vaccination.

## Figures and Tables

**Figure 1 vaccines-12-01202-f001:**
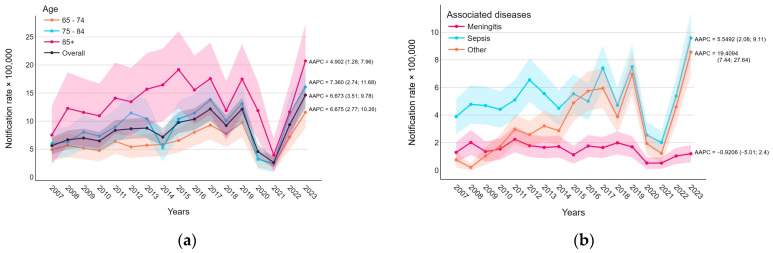
Notification rates of invasive pneumococcal disease (per 100,000 inhabitants) trends with average annual percentage change (AAPC) and 95% confidence intervals: (**a**) by age group and (**b**) by invasive bacterial diseases.

**Figure 2 vaccines-12-01202-f002:**
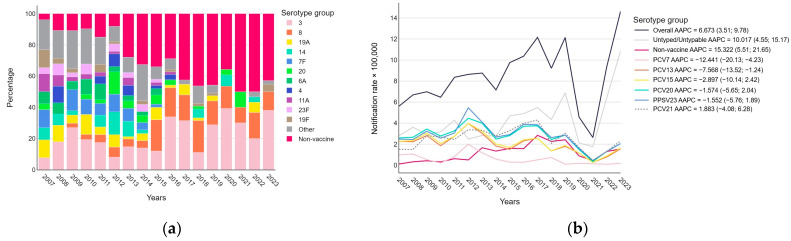
*Streptococcus pneumoniae* serotype distribution over the study years excluding untyped cases (**a**) and notification rates of invasive pneumococcal disease (per 100,000 inhabitants) trends with average annual percentage change (AAPC) and 95% confidence intervals by vaccine group (**b**).

**Table 1 vaccines-12-01202-t001:** Characteristics of invasive pneumococcal disease notifications by age group.

Variable		Age			Total (N = 1527)	*p*-Value
		65–74 (N = 582)	75–84 (N = 588)	85+ (N = 357)		
Sex	Female	229 (39.3%)	243 (41.3%)	212 (59.4%)	684 (44.8%)	<0.0001
	Male	353 (60.7%)	345 (58.7%)	145 (40.6%)	843 (55.2%)	
Notification Season	Winter	229 (39.3%)	247 (42.0%)	133 (37.3%)	609 (39.9%)	0.7721
	Spring	141 (24.2%)	139 (23.6%)	83 (23.2%)	363 (23.8%)	
	Summer	44 (7.6%)	40 (6.8%)	31 (8.7%)	115 (7.5%)	
	Autumn	168 (28.9%)	162 (27.6%)	110 (30.8%)	440 (28.8%)	
Associated Diseases						
	Meningitis	156 (27.0%)	90 (15.6%)	16 (4.6%)	262 (17.4%)	<0.0001
	NA	4	10	7	21	
	Sepsis	364 (63.2%)	354 (61.2%)	227 (64.9%)	945 (62.8%)	0.5300
	NA	6	10	7	23	
	Other	215 (41.1%)	260 (49.6%)	168 (52.8%)	643 (47.1%)	0.0015
	NA	59	64	39	162	
Material						
	Blood	412 (96.0%)	473 (95.0%)	316 (96.9%)	1201 (95.8%)	0.3780
	NA	153	90	31	274	
	Liquor	125 (46.5%)	68 (29.6%)	13 (11.6%)	206 (33.7%)	<0.0001
	NA	313	358	245	916	
	Other	7 (5.3%)	10 (6.8%)	4 (4.8%)	21 (5.8%)	0.8439
	NA	450	442	274	1166	
Untyped/Untypable Serotype		306 (52.6%)	315 (53.6%)	173 (48.5%)	794 (52.0%)	0.2934
Serotype	11A	5 (1.8%)	8 (2.9%)	1 (0.5%)	14 (1.9%)	0.2499
	14	10 (3.6%)	10 (3.7%)	10 (5.4%)	30 (4.1%)	
	19A	12 (4.3%)	18 (6.6%)	7 (3.8%)	37 (5.0%)	
	19F	8 (2.9%)	4 (1.5%)	2 (1.1%)	14 (1.9%)	
	20	5 (1.8%)	11 (4.0%)	9 (4.9%)	25 (3.4%)	
	23F	7 (2.5%)	3 (1.1%)	4 (2.2%)	14 (1.9%)	
	3	53 (19.2%)	73 (26.7%)	32 (17.4%)	158 (21.6%)	
	4	4 (1.4%)	6 (2.2%)	7 (3.8%)	17 (2.3%)	
	6A	10 (3.6%)	5 (1.8%)	5 (2.7%)	20 (2.7%)	
	7F	12 (4.3%)	8 (2.9%)	9 (4.9%)	29 (4.0%)	
	8	34 (12.3%)	24 (8.8%)	23 (12.5%)	81 (11.1%)	
	Non-vaccine	86 (31.2%)	77 (28.2%)	55 (29.9%)	218 (29.7%)	
	Other	30 (10.9%)	26 (9.5%)	20 (10.9%)	76 (10.4%)	
	NA	306	315	173	794	
Vaccinated *	Yes	101 (35.2%)	88 (32.7%)	29 (16.9%)	218 (29.9%)	0.0001
	NA	295	319	185	799	
Sequelae	Yes	8 (1.6%)	7 (1.4%)	0 (0%)	15 (1.2%)	0.0592
	NA	81	96	53	230	
Auditory Sequelae	Yes	1 (0.2%)	2 (0.4%)	0 (0%)	3 (0.2%)	0.6266
	NA	109	116	59	284	
Neurological Sequelae	Yes	3 (0.6%)	2 (0.4%)	0 (0%)	5 (0.4%)	0.4590
	NA	107	115	59	281	
Amputation	Yes	0 (0%)	1 (0.2%)	0 (0%)	1 (0.1%)	0.6188
	NA	107	115	59	281	
Other Sequelae	Yes	5 (1.1%)	3 (0.6%)	0 (0%)	8 (0.6%)	0.2074
	NA	107	116	59	282	
Deceased (CFR%)		53 (9.1%)	73 (12.4%)	61 (17.1%)	187 (12.2%)	0.0014

Legend: NA, not available. * Vaccination status not available for records prior to 2012.

## Data Availability

The data that support the findings of this study are available on request from the corresponding author.

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
