# Peer review of "Invasive Pneumococcal Diseases in People over 65 in Veneto Region Surveillance"

_vaccines, 2024, doi:10.3390/vaccines12111202_

Round 1
Reviewer 1 Report
Comments and Suggestions for Authors
The study by Cocchio et al. reported IPD in elderly people from Veneto Region. The study is interesting, include 1527 cases of IPD, and the authors highlight that serotypes 7F, 6A, 4, 11A, and 23F have significantly decreased and have disappeared since 2016, and , except for serotype 11A, which is included in PCV20 and PPSV23, all the other serotypes have been covered by vaccines in use for many years.
I have only minor comments:
1.- There are some typo in the name of the microorganisms
2.- The fact that in more than half of the population there are not data about vaccination may limit some conclusions about the effect of pneumococcal vaccines with the reported serotypes.
3.- Add the meaning of "NA" in legend of table 1
Comments on the Quality of English Language
The quality of the English if ok, there are some typos in the names of the microorganism, such in line 189
Author Response
Comment: There are some typo in the name of the microorganisms.
Answer: We checked names of the microorganisms and corrected the typos.
Comment: The fact that in more than half of the population there are not data about vaccination may limit some conclusions about the effect of pneumococcal vaccines with the reported serotypes.
Answer: We recognise this limitation, we have emphasized and clarified this aspect in the discussion.
Comment: Add the meaning of "NA" in legend of table 1.
Answer: Thank you, done.
Comment: The quality of the English if ok, there are some typos in the names of the microorganism, such in line 189.
Answer: Thank you, done.
Reviewer 2 Report
Comments and Suggestions for Authors
I was invited to revise the paper entitled "Invasive Pneumococcal Diseases in People Over 65 in Veneto Region Surveillance". It was a cohort study from a Northern Italian Region, aimed to evaluate the incidence of IPD among patients aged over 65 years.
The topic is relevant for public health and it can improve the knowledge on this field.
Strenghts:
- Long study period;
- Strong methodology that can be replicated;
- Well written manuscript.
Observations:
- Is the incidence rate standardized by age and gender?
- The impact of PCV20, introduced in Italy only during last years could decrease the trend in next years. Authors should discuss this point;
- About jointpoint, Authors should add, as supplementary material, the sub-analysis of each joint-point reporting the related APC. It can improve the understanding of trend variations across the study period;
- Authors should compare these results with similar study performed in Italy during last years.
Author Response
Comment: Is the incidence rate standardized by age and gender?
Answer: No, the incidence rates were not directly standardized by age and sex, as the focus of the study was specifically on the elderly population in the Veneto region. The goal was to obtain age-specific rates and evaluate their trends over time. For each year and age group, the incidence rate was calculated using the resident population in Veneto within the corresponding age group and year as the denominator. By concentrating on a narrow segment of the population and using small age classes, standardization would have yielded similar results, as the demographic characteristics of the over-65 population in Veneto have remained relatively stable in terms of age and sex over the study years.
Comment: The impact of PCV20, introduced in Italy only during last years could decrease the trend in next years. Authors should discuss this point;
Answer: Thank you for the suggestion. We have highlighted this point in the section on future perspectives.
Comment: About jointpoint, Authors should add, as supplementary material, the sub-analysis of each joint-point reporting the related APC. It can improve the understanding of trend variations across the study period;
Answer: Thank you, done.
Comment: Authors should compare these results with similar study performed in Italy during last years.
Answer: Thank you for the suggestion. We have clarified this point further in the discussion when comparing our data with similar Italian studies, and we have also included a specific reference to a prevalence study on community-acquired pneumonia.